# Sense of coherence and risk of breast cancer

Kejia Hu[1], Mikael Eriksson[2], Yvonne Wengström[3], Kamila Czene[2], Per Hall[2], Fang Fang[1]*

[1]Unit of Integrative Epidemiology, Institute of Environmental Medicine, Karolinska Institutet, Stockholm, Sweden; [2]Department of Medical Epidemiology and Biostatistics, Karolinska Institutet, Stockholm, Sweden; [3]Department of Neurobiology, Care Sciences and Society, Karolinska Institutet & Theme Cancer, Karolinska University Hospital, Stockholm, Sweden

**Abstract** Sense of coherence (SoC) is the origin of health according to Antonovsky. The link between SoC and risk of cancer has however rarely been assessed. We performed a cohort study of 46,436 women from the Karolinska Mammography Project for Risk Prediction of Breast Cancer (Karma). Participants answered a SoC-13 questionnaire at recruitment to Karma and were subsequently followed up for incident breast cancer. Multivariate Cox models were used to assess the hazard ratios (HRs) of breast cancer in relation to SoC. We identified 771 incident cases of breast cancer during follow-up (median time: 5.2 years). No association was found between SoC, either as a categorical (strong vs. weak SoC, HR: 1.08, 95% CI: 0.90–1.29) or continuous (HR: 1.08; 95% CI: 1.00–1.17 per standard deviation increase of SoC) variable, and risk of breast cancer. In summary, we found little evidence to support an association between SoC and risk of breast cancer.

## Introduction

Health, by definition of the World Health Organization (WHO), is a state of complete physical, mental and social well-being (*World Health Organization, 2014*). The theory of salutogenesis was developed by *Antonovsky, 1979* to study factors that support human health and well-being, in contrast to 'pathogenesis' which studies factors that cause diseases, with a specific focus on understanding the relationship between health, stress, and coping. According to this theory, health is a lifelong and dynamic process of actively coping with a variety of stress factors one might encounter in life (i.e., stimuli), and, although stress is ubiquitous, people experience different health outcomes in response to stress.

Sense of coherence (SoC) is the essence of the theory of salutogenesis, defined as "a global orientation that expresses the extent to which one has a pervasive, enduring though dynamic feeling of confidence that (1) the stimuli deriving from one's internal and external environments in the course of living are structured, predictable, and explicable; (2) the resources are available to one to meet the demands posed by these stimuli; and (3) these demands are challenges, worthy of investment and engagement' (*Antonovsky, 1987*). Accordingly, SoC has three components, namely comprehensibility, manageability and meaningfulness, and salutogenesis is greatly dependent on a strong SoC. Generalized Resistance Resources are one of the key determinants for SoC, consisting all resources that help a person cope and are effective in avoiding or combating a range of stress factors, including socioeconomic standing and social support (*Antonovsky, 1972*; *Idan et al., 2016*).

The theory of salutogenesis has been applied in different fields of health and medicine. For instance, a potential link has been suggested between SoC and different somatic diseases (*Flensborg-Madsen et al., 2005*), including diabetes (*Eriksson et al., 2013*) and cardiovascular diseases

*For correspondence:
fang.fang@ki.se

**Competing interests:** The authors declare that no competing interests exist.

(*Poppius et al., 1999*; *Nasermoaddeli et al., 2004*). Yet, little is known about the link between SoC and cancer development. Animal research has proposed stress as a contributing factor to the initiation, growth, and metastasis of cancer (*Lutgendorf and Andersen, 2015*). In addition, human mechanistic studies have suggested various pathogenic processes (e.g., altered antiviral defenses, DNA repair, and cellular aging) as potential pathways linking together stress and oncogenesis (*Antoni et al., 2006*). However, evidence from previous studies on the association of stress with cancer incidence in humans remains largely equivocal (*Chida et al., 2008*). One reason for the observed diverging results might be the highly heterogeneous definitions and measurements of stress of the existing literature (*Fang et al., 2015*). In contrary, there is so far one study, to the best of our knowledge, that assessed the role of SoC on cancer development, and found an association between weak SoC and an increased risk of cancer during an 8 year follow-up, although not 12 year follow-up, of a population of middle-aged Finnish men (*Poppius et al., 2006*). No study has addressed this research question among women and for female cancer. To this end, we took advantage of a mammography screening cohort in Sweden and assessed the association of SoC with the subsequent risk of breast cancer.

## Results

In total, we included 46,436 eligible women (65.5% of all Karma participants) in the present study. The median age at baseline was 54 years (range: 40–74). More than half of the women were postmenopausal, and had college education or above (*Table 1*). Women with strong SoC were slightly older, more educated, less likely to smoke, and more physically active.

In the linear regression, we found SoC to be positively associated with age (coefficient = 0.13, 95% CI: 0.12 to 0.14), number of births (coefficient = 0.49, 95% CI: 0.39 to 0.59), and a higher physical activity (coefficient = 1.24, 95% CI: 0.98 to 1.5 for high vs. low physical activities) (*Figure 1*). Higher BMI (coefficient = −0.17, 95% CI: −0.19 to −0.14), non-European ancestry (coefficient = −5.15, 95% CI: −5.8 to −4.5), less education (coefficient = −2.13, 95% CI: −2.46 to −1.8 for ≤10 years vs. >12 years of education), previous or current smoking (coefficient = −0.99, 95% CI: −1.22 to −0.77 for previous smoker vs. non-smoker; coefficient = −2.45, 95% CI: −2.78 to −2.12 for current smoker vs. non-smoker), and never or ≥10 g/d drinking (coefficient = −2.23, 95% CI: −2.51 to −1.95 for non-drinker vs. <10 g/d of alcohol consumption; coefficient = −0.63, 95% CI: −0.9 to −0.36 for ≥10 g/d vs. <10 g/d of alcohol consumption) were all associated with a lower SoC score.

We identified 771 incident cases of breast cancer during follow-up (median time of follow-up: 5.2 years; range: 0.03–7 years). We did not observe an association between SoC and risk of breast cancer, when comparing women with strong SoC to women with weak SoC (HR: 1.08, 95% CI: 0.90–1.29), with or without multivariable adjustment (*Table 2*). No statistically significant association was noted either when using SoC as a continuous variable (HR: 1.08, 95% CI: 1.00–1.17 per SD increase of SoC). Sensitivity analysis using flexible parametric model demonstrated a constantly null association between SoC and risk of breast cancer (*Figure 2*). A null association was seen for all subtypes of breast cancer (*Table 3*).

## Discussion

We found little evidence to support an association between SoC and the risk of breast cancer in a population-based cohort study, including 46,436 women at 40–74 years who attended mammography screening in Sweden. Our study represents the first attempt to date to examine the association of SoC with the subsequent risk of breast cancer.

The correlations noted between SoC and different characteristics of the women are consistent with results from previous studies, which found SoC to increase with age (*Poppius et al., 2006*; *Nilsson et al., 2010*), and decrease with an unhealthy lifestyle (*Igna et al., 2008*; *Midanik et al., 1992*). We also found SoC to increase with number of births and years of education, which has not been reported previously. SoC was found to be lower among women with non-European ancestry, which is in line with a previous study reporting a lower SoC for immigrants than their local counterparts (*Erim et al., 2011*), perhaps due to the challenge in sociocultural and economic adaptation among the immigrants. Finally, the internal consistency of SoC (Cronbach's Alpha = 0.87) noted in

**Table 1.** Characteristics of study participants by sense of coherence at baseline.

| Characteristics | Weak SoC [*] | Moderate SoC | Strong SoC |
|---|---|---|---|
| No. | 16,066 | 16,099 | 14,271 |
| Age at baseline, mean [SD] | 54.0 [9.7] | 55.1 [9.8] | 55.6 [9.9] |
| European ancestry (%) | 15,341 (95.5) | 15,726 (97.7) | 13,970 (97.9) |
| Education, years (%) | | | |
| ≤10 | 2412 (15.0) | 2143 (13.3) | 1885 (13.2) |
| 12 | 5500 (34.2) | 4776 (29.7) | 3744 (26.2) |
| >12 | 7767 (48.3) | 8829 (54.8) | 8330 (58.4) |
| Body mass index (Kg/m$^2$), mean [SD] | 25.6 [4.5] | 25.1 [4.1] | 25.0 [4.0] |
| Smoking (%) | | | |
| Never | 7028 (43.7) | 7663 (47.6) | 7285 (51.0) |
| Previous | 6569 (40.9) | 6481 (40.3) | 5535 (38.8) |
| Current | 2430 (15.1) | 1927 (12.0) | 1430 (10.0) |
| Alcohol consumption (%) | | | |
| Non-drinker | 3628 (22.6) | 2700 (16.8) | 2373 (16.6) |
| <10 g/d | 9121 (56.8) | 10,013 (62.2) | 9136 (64.0) |
| ≥10 g/d | 3168 (19.7) | 3259 (20.2) | 2681 (18.8) |
| Physical activity (%) | | | |
| Low | 5926 (36.9) | 5371 (33.4) | 4486 (31.4) |
| Medium | 5077 (31.6) | 5489 (34.1) | 4947 (34.7) |
| High | 4873 (30.3) | 5146 (32.0) | 4769 (33.4) |
| Age at menarche, mean [SD] | 13.1 [1.5] | 13.1 [1.5] | 13.1 [1.4] |
| Age at first birth, mean [SD] | 27.0 [5.5] | 27.2 [5.2] | 27.2 [5.1] |
| No. of pregnancies, median (range) | 2 (0–15) | 2 (0–13) | 2 (0–15) |
| No. of births, median (range) | 2 (0–11) | 2 (0–11) | 2 (0–11) |
| Ever use of oral contraceptives (%) | 13,429 (83.6) | 13,549 (84.2) | 12,018 (84.2) |
| Ever use of HRT (%) | 4407 (27.4) | 4362 (27.1) | 3752 (26.3) |
| Post-menopausal (%) | 8590 (53.5) | 9294 (57.7) | 8476 (59.4) |
| BC in first degree relatives (%) | 2002 (12.5) | 2006 (12.5) | 1821 (12.8) |
| Benign breast disorders (%) | 3653 (22.7) | 3659 (22.7) | 3143 (22.0) |
| Other malignancies (%) | 856 (5.3) | 882 (5.5) | 799 (5.6) |
| Breast density[†], mean [SD] | 22.0 [19.5] | 22.4 [19.7] | 22.1 [19.3] |

Abbreviations: BC, breast cancer; HRT, hormone receptor therapy; SD, standard deviation; SoC, sense of coherence.

[*]: SoC was categorized according to tertiles of the total SoC score measured at baseline for all women.

[†]: Breast density (%) was calculated using absolute dense area (cm$^2$) divided by total breast area (cm$^2$) using STRA-TUS software.

the present study is also in line with previous studies (Cronbach's Alpha ranged from 0.70 to 0.92) (*Eriksson and Lindström, 2005*).

To our knowledge, few studies have examined the link between SoC and risk of cancer. *Poppius et al., 2006* reported a 52% higher risk of any cancer during an 8 year follow-up of middle-aged men in Finland, when comparing men with a weak SoC to men with a strong SoC. The risk increment diminished however when expanding the follow-up to 12 years, although a statistically significantly increased cancer risk was observed among older men (i.e., above 55 years of age at baseline) regardless of the length of follow-up (*Poppius et al., 2006*). Our study is the first to examine the role of SoC on cancer risk among women and showed a null association between SoC and the risk of breast cancer development. The different study population (men versus women) and cancer

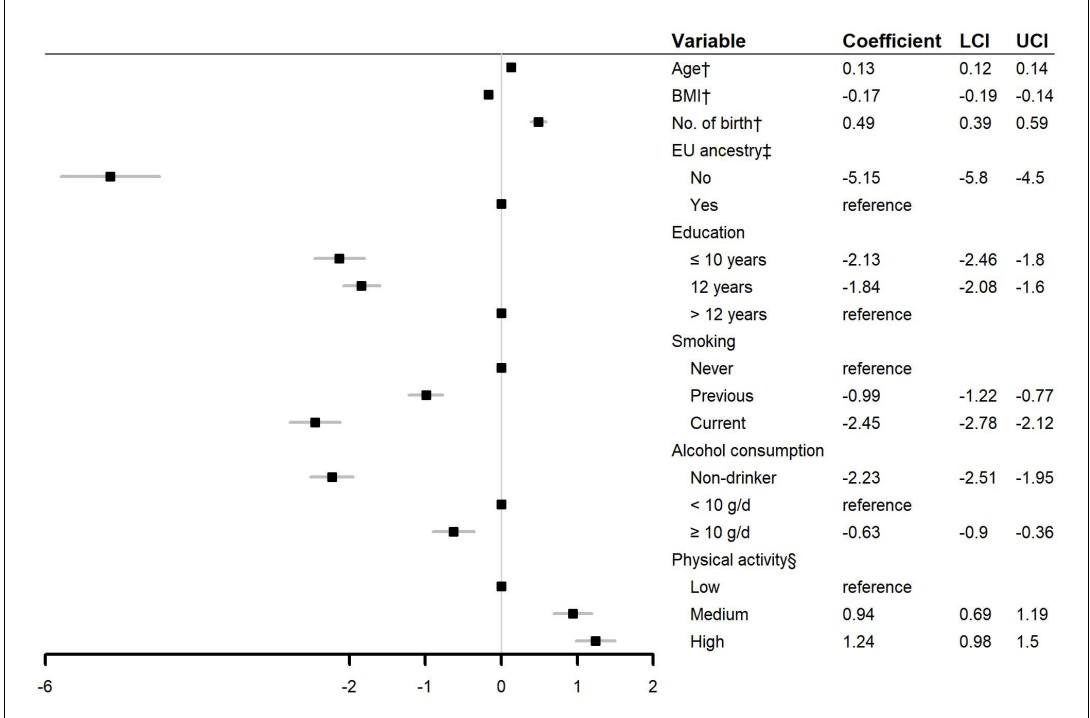

| Variable | Coefficient | LCI | UCI |
|---|---|---|---|
| Age† | 0.13 | 0.12 | 0.14 |
| BMI† | -0.17 | -0.19 | -0.14 |
| No. of birth† | 0.49 | 0.39 | 0.59 |
| EU ancestry‡ | | | |
| No | -5.15 | -5.8 | -4.5 |
| Yes | reference | | |
| Education | | | |
| ≤ 10 years | -2.13 | -2.46 | -1.8 |
| 12 years | -1.84 | -2.08 | -1.6 |
| > 12 years | reference | | |
| Smoking | | | |
| Never | reference | | |
| Previous | -0.99 | -1.22 | -0.77 |
| Current | -2.45 | -2.78 | -2.12 |
| Alcohol consumption | | | |
| Non-drinker | -2.23 | -2.51 | -1.95 |
| < 10 g/d | reference | | |
| ≥ 10 g/d | -0.63 | -0.9 | -0.36 |
| Physical activity§ | | | |
| Low | reference | | |
| Medium | 0.94 | 0.69 | 1.19 |
| High | 1.24 | 0.98 | 1.5 |

**Figure 1.** Linear associations between sociodemographic and lifestyle factors and sense of coherence. Abbreviations: BMI: body mass index; LCI: lower boundary of 95% confidence interval; UCI, upper boundary of 95% confidence interval. †: Continuous variable. ‡ European ancestry was defined as if the participant and her parents were all born in Europe. § Physical activity was categorized by tertiles of the metabolic equivalent of task of daily physical activity during the last month.

outcomes (all cancers among men versus female breast cancer) might have contributed to the diverging results observed between Poppius et al and the present study. The role of SoC in other cancer types among women, in populations outside the Nordic countries, or among other age groups needs to be further investigated. The lack of association between SoC and risk of breast cancer does however not necessarily exclude a possible link between SoC and prognosis of breast

**Table 2.** Hazard ratios and 95% confidence intervals of breast cancer in relation to sense of coherence measured at baseline.

| SoC | 1000 PYs | Event (IR) | Model 1[*] | Model 2[†] |
|---|---|---|---|---|
| In three categories[‡] | | | | |
| Weak SoC | 83 | 251 (3.0) | 1.00 | 1.00 |
| Moderate SoC | 83 | 264 (3.2) | 1.05 (0.88–1.25) | 1.00 (0.84–1.19) |
| Strong SoC | 74 | 256 (3.5) | 1.15 (0.97–1.37) | 1.08 (0.90–1.29) |
| Per SD increase[§] | 240 | 771 (3.2) | 1.11 (1.03–1.20) | 1.08 (1.00–1.17) |

Abbreviations: PYs, person-years; IR, incidence rate per 1000 person-years; SoC, sense of coherence; SD, standard deviation.

[*]: Estimates were not adjusted for any covariate. [†]: Estimated were adjusted for sociodemographic factors (age, ancestry, education) and lifestyle factors (body mass index, smoking, alcohol consumption, physical activity), and known risk factors for breast cancer including reproductive and hormonal factors (age at menarche, age at first birth, No. of pregnancies, No. of births, use of contraceptives, use of hormone replacement therapy, menopausal status), as well as family history of breast cancer, benign breast disorders, other malignancies, and breast density.

[‡]: Total SoC score at baseline was calculated and categorized as weak, moderate, and strong according to tertile distribution of all women. [§]: SoC was treated as a continuous variable after standardization using z-score method.

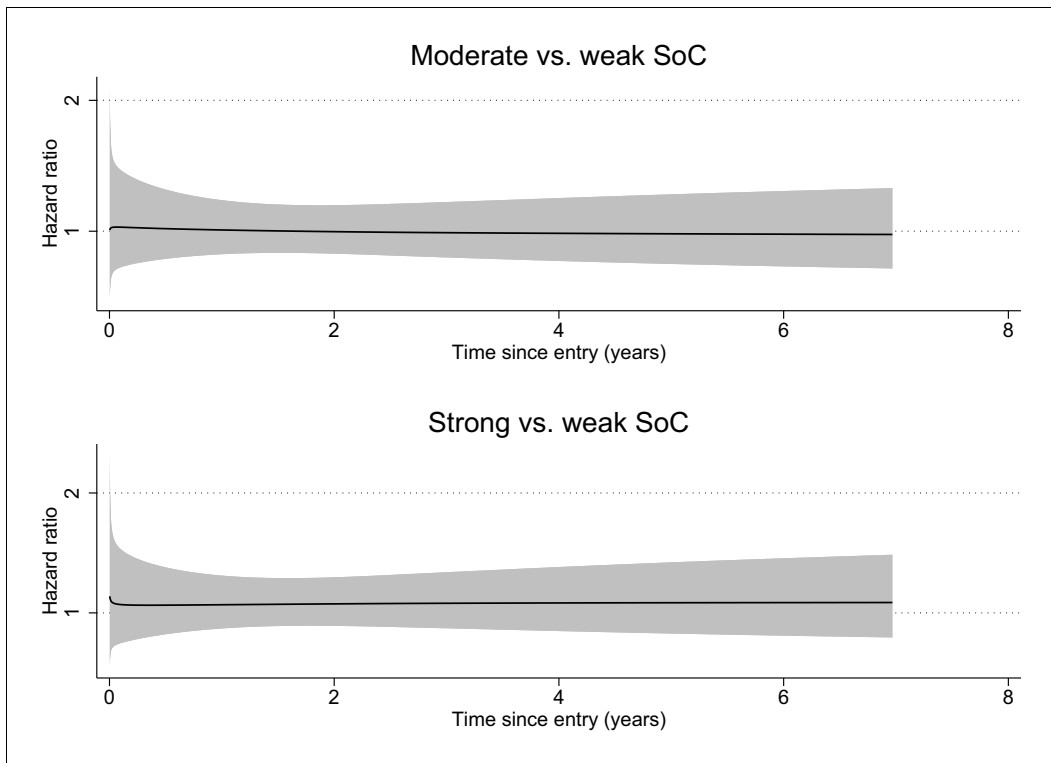

**Figure 2.** Time-dependent effect of sense of coherence on incident breast cancer. Abbreviations: SoC, sense of coherence. Time-varying hazard ratios of incident breast cancer for moderate and strong SoC compared to weak SoC were derived from a flexible parametric model, adjusted for sociodemographic factor (age, ancestry, education), lifestyle factors (body mass index, smoking, alcohol consumption, physical activity), and known risk factors for breast cancer including reproductive and hormonal factors (age at menarche, age at first birth, No. of pregnancies, No. of births, use of contraceptives, use of hormone replacement therapy, menopausal status), as well as family history of breast cancer, benign breast disorders, other malignancies, and breast density.

The online version of this article includes the following source data for figure 2:

**Source data 1.** Summary data for *Figure 2*.

cancer. For instance, a previous study has found higher SoC to be associated with a decreased risk of both breast cancer-specific mortality and all-cause mortality (*Lindblad et al., 2018*).

The strength of the present study includes the large-scale population-based cohort design, the comprehensive information collected from questionnaires, and the complete follow-up through link-ages to national registers, which reduce greatly the possibility of systematic and random errors. There are however also limitations in our study. In the SoC-13 questionnaire used in Karma, 12 out of the total 13 items were reversed as a result of restructuring answers. Reverse item bias may there-fore exist in SoC scoring due to acquiescence, careless responding, or confirmation bias (*Weijters et al., 2013*). However, this concern may be partly alleviated by the high internal consis-tency noted between the items. Our study used the National Cancer Register and INCA to ascertain incident cases of breast cancer during follow-up. As the accuracy and completeness of breast cancer registration have both been shown as excellent in the National Cancer Register (underreporting rate of 1.1% for breast cancer in women under 70 years) (*Barlow et al., 2009*) and INCA (underreporting rate of 0.1% compared to the National Cancer Register) (*Löfgren et al., 2019*), the present results are unlikely greatly affected by misclassification of outcome measurement. Similarly, the Total Popu-lation Register (*Ludvigsson et al., 2016*) and Causes of Death Register (*Brooke et al., 2017*) have highly complete information on migration and death, largely alleviating concern about loss to follow-up. Karma is a mammography screening cohort, with the participants actively attending mammogra-phy screening and willing to participate in the research cohort (*Gabrielson et al., 2017*). In the pres-ent study, we only included Karma participants who have responded to SoC-13. Consequently, the participants included in the final analysis might differ from the entire female population in Sweden,

**Table 3.** Hazard ratios and 95% confidence intervals of breast cancer subtypes in relation to sense of coherence measured at baseline.

| Subtypes* | Weak SoC† | | | Moderate SoC | | | Strong SoC | | |
|---|---|---|---|---|---|---|---|---|---|
| | 1000 PYs | Event (IR) | HR | 1000 PYs | Event (IR) | HR (95% CI)‡ | 1000 PYs | Event (IR) | HR (95% CI)‡ |
| Luminal A | 83 | 105 (1.3) | Ref. | 83 | 99 (1.2) | 0.91 (0.69–1.20) | 74 | 100 (1.4) | 1.05 (0.79–1.39) |
| Luminal B1 | 83 | 17 (0.2) | Ref. | 83 | 22 (0.3) | 1.23 (0.65–2.34) | 74 | 12 (0.2) | 0.63 (0.30–1.34) |
| Luminal B2 | 83 | 12 (0.1) | Ref. | 83 | 14 (0.2) | 1.28 (0.58–2.80) | 74 | 15 (0.2) | 1.47 (0.67–3.23) |
| Triple-negative | 83 | 9 (0.1) | Ref. | 83 | 15 (0.2) | 1.52 (0.66–3.50) | 74 | 6 (0.1) | 0.71 (0.24–2.04) |
| HER2-enriched | 83 | 9 (0.1) | Ref. | 83 | 8 (0.1) | 0.78 (0.30–2.06) | 74 | 7 (0.1) | 0.82 (0.30–2.28) |
| Unclassified | 83 | 99 (1.2) | Ref. | 83 | 106 (1.3) | 0.99 (0.75–1.31) | 74 | 116 (1.6) | 1.23 (0.93–1.62) |

Abbreviations: PYs, person-years; IR, incidence rate per 1000 person-years; SoC, sense of coherence.

*: Breast cancer subtypes were classified based on estrogen receptor (ER), progesterone receptor (PR) and human epidermal growth factor receptor 2 (HER2). Luminal A: ER+, PR+, HER2-; Luminal B1: ER+, PR-, HER2-; Luminal B2: ER+, HER2+; Triple-negative: ER-, PR-, HER2-; HER2-enriched: ER-, PR-, HER2+; the rest were unclassified, mostly because of undetermined HER2 status.

†: A total SoC score at baseline was calculated and categorized according to tertiles in all women: weak, moderate, and strong. ‡: Estimates were adjusted for sociodemographic factor (age, ancestry, education), lifestyle factors (body mass index, smoking, alcohol consumption, physical activity), and known risk factors for breast cancer including reproductive and hormonal factors (age at menarche, age at first birth, No. of pregnancies, No. of births, use of contraceptives, use of hormone replacement therapy, menopausal status), as well as family history of breast cancer, benign breast disorders, other malignancies, and breast density.

for instance, in terms of socioeconomic status. As a result, caution is needed when extrapolating these results to all women in Sweden or to female populations outside Sweden.

In summary, based on a large cohort of Swedish women, we found little evidence to support an association between SoC and the risk of breast cancer.

## Materials and methods

### Study design

The Karolinska Mammography Project for Risk Prediction of Breast Cancer (Karma) cohort recruited 70,872 women who attended a mammography screening or clinical mammography between October 2010 and March 2013 at any of the four mammography units (Stockholm South General Hospital, Helsingborg Hospital, Landskrona Hospital, and Skåne University Hospital, Lund) in Sweden (*Gabrielson et al., 2017*). Participants answered a comprehensive questionnaire, including SoC-13 (*Antonovsky, 1993*), at the baseline visit (recruitment to Karma). More information about Karma, including details of the questionnaires, is available at karmastudy.org. In this study, we included participants who had no prevalent breast cancer, were at the age of 40 to 74 years, and had completed at least two-thirds of the items included in SoC-13 questionnaire at the baseline visit. The participants were then individually followed up from the date of baseline visit, through cross-linkages to various Swedish national registers, using the unique Swedish personal identity numbers. The personal identity numbers were replaced with study specific identity numbers at the Swedish National Board of Health and Welfare before data were received by the researchers. The follow-up ended at a diagnosis of breast cancer (through INCA and National Cancer Register), emigration out of Sweden (through Total Population Register), death (through Causes of Death Register), or October 6th, 2017, whichever occurred first.

### Sense of coherence (SoC)

SoC was measured at baseline using the SoC-13 questionnaire (*Supplementary file 1*), which was developed by *Antonovsky, 1993* and validated in different studies (*Eriksson and Mittelmark, 2016*; *Holmefur et al., 2015*). A total score of SoC was calculated by the sum of included items, with the scores of a few items reversed as needed. As SOC-13 has high internal consistency (*Eriksson and Mittelmark, 2016*) (Cronbach's alpha = 0.87 in our study), any missing value of a single item was replaced by the mean value of the other items if at least two-thirds of the items were

answered by one single person (*Volanen, 2011*). We categorized the total score of SoC as 'weak', 'moderate', or 'strong' according to the tertile distribution. We also used the total score as a continuous variable after z-score standardization and estimated effect per standard deviation (SD) increase.

## Other covariables

In this study, we used information from the baseline questionnaire, including sociodemographic factors (age, ancestry, education), lifestyle factors (body mass index [BMI], smoking, drinking, physical activity), and other known risk factors for breast cancer including reproductive and hormonal factors, family history of breast cancer, history of benign breast disorders or other malignancies. We also used breast density percent measured at baseline by previously reported STRATUS software (*Eriksson et al., 2018*). Finally, to study different subtypes of breast cancer, we also obtained information on the tumor markers for women with incident breast cancer identified during follow-up. For example, estrogen receptor (ER), progesterone receptor (PR), and human epidermal growth factor receptor 2 (HER2) were obtained through linkage to INCA.

## Statistical analysis

We first described the baseline characteristics of women according to different categories of SoC (weak, moderate, or strong). We assessed the correlations between sociodemographic and lifestyle factors and SoC using linear regression models. We then used Cox proportional hazards models to assess the association between SoC and the risk of breast cancer. We first fitted a model not adjusted for any other covariables (Model 1). In a full model (Model 2), we adjusted the analyses for sociodemographic factors (age, ancestry, and education) and lifestyle factors (body mass index, smoking, alcohol consumption, and physical activity), and known risk factors for breast cancer including reproductive and hormonal factors (age at menarche, age at first birth, No. of pregnancies, No. of births, use of contraceptives, use of hormone replacement therapy, menopausal status), as well as family history of breast cancer, benign breast disorders, other malignancies, and breast density. We tested proportional hazard assumptions of the Cox models using Schoenfeld residuals and found no major deviation from the assumptions. To assess the potential temporal pattern of the association, we did a sensitivity analysis by fitting a flexible parametric model allowing for the effect of SoC on breast cancer risk to vary over time since study entry. We further performed a sub-analysis to assess whether the studied association would differ between subtypes of breast cancer using the full model. The statistical analyses were performed using Stata, version 16, StataCorp LP and R (version 4.0.2). We used a two-sided $p<0.05$ to indicate statistical significance.

## Acknowledgements

This study was supported by the Swedish Cancer Society (grant number: CAN 2017/322) and the Swedish Research Council for Health, Working Life and Welfare (grant number: 2017–00531); by Karolinska Institutet Senior Researcher Award and Strategic Research Area in Epidemiology Award; and by China Scholarship Council (No. 201806240005). We thank the participants and the employees at Karma, Unilabs and others, Stockholm South General Hospital, Helsingborg Hospital, Landskrona Hospital and Skåne University Hospital, Lund for their contribution to the work.

## Additional information

### Funding

| Funder | Grant reference number | Author |
| --- | --- | --- |
| Cancerfonden | CAN 2017/322 | Fang Fang |
| Swedish Research Council for Health, Working Life and Welfare | 2017-00531 | Fang Fang |
| Karolinska Institutet | Senior Researcher Award | Fang Fang |
| Karolinska Institutet | Strategic Research Area in | Fang Fang |

Epidemiology Award

| China Scholarship Council | 201806240005 | Kejia Hu |
| --- | --- | --- |

The funders had no role in study design, data collection and interpretation, or the decision to submit the work for publication.

### Author contributions
Kejia Hu, Software, Formal analysis, Funding acquisition, Visualization, Methodology, Writing - original draft, Writing - review and editing; Mikael Eriksson, Resources, Data curation, Software, Investigation, Writing - review and editing; Yvonne Wengström, Supervision, Validation, Writing - review and editing; Kamila Czene, Resources, Investigation, Methodology, Writing - review and editing; Per Hall, Conceptualization, Resources, Supervision, Investigation, Project administration, Writing - review and editing; Fang Fang, Conceptualization, Supervision, Funding acquisition, Methodology, Writing - original draft, Project administration, Writing - review and editing

### Author ORCIDs
Kejia Hu (iD) https://orcid.org/0000-0001-6680-8107
Fang Fang (iD) https://orcid.org/0000-0002-3310-6456

### Ethics
Human subjects: All procedures performed in studies involving human participants were in accordance with the ethical standards of the institutional and national research committee and with the 1964 Helsinki declaration and its later amendments or comparable ethical standards. Approval was granted by the Regional Ethics Review Board in Stockholm, Sweden (Dnr 2010/958-31/1). Informed consent was obtained from all individual participants included in the study.

### Decision letter and Author response
Decision letter https://doi.org/10.7554/eLife.61469.sa1
Author response https://doi.org/10.7554/eLife.61469.sa2

## Additional files

### Supplementary files
• Source code 1. Stata script for *Table 2* and *Figure 2*.

• Supplementary file 1. SoC-13 questionnaire of sense of coherence used in the Karolinska Mammography Project for Risk Prediction of Breast Cancer (Karma) cohort. The following questions were asked to investigate how the participants experienced these situations. A 7-point Likert-type scale is used to answer each question, with one corresponding to 'very seldom or never', and seven corresponding to 'very often'. To simplify the structure, the answer of item 4 to 9, 11 to 13 have been reversed on the basis of the original SoC-13 questionnaire.

• Transparent reporting form

### Data availability
The datasets analysed during the present study can be shared and are available from the corresponding author on reasonable request. More information regarding the data access to KARMA can be found at: https://karmastudy.org/contact/data-access/. The data are not publicly available due to Swedish laws.

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
