## [Decision Letter]

**Acceptance summary:**

Sense of coherence refers to a person's psychological ability to properly adapt to stress and adversity during the life course. The scale was developed to measure how people view life and to identify how a person may use her/ his intrinsic resources to overcome difficulties while maintaining their health. It seeks to understand why some people remain healthy under stress while others become sick. Using the excellent database from the Karolinska Mammography Project for Risk of Breast Cancer, the current study did not show an association between sense of coherence and breast cancer.

**Decision letter after peer review:**

Thank you for submitting your article "Sense of coherence and risk of breast cancer" for consideration by *eLife*. Your article has been reviewed by three peer reviewers, one of whom is a member of our Board of Reviewing Editors, and the evaluation has been overseen by a Senior Editor. The following individual involved in review of your submission has agreed to reveal their identity: Jitka Pikhartova (Reviewer #3).

As is customary in *eLife*, the reviewers have discussed their critiques with one another. What follows below is the Reviewing Editor's edited compilation of the essential and ancillary points provided by reviewers in their critiques and in their interaction post-review. Please submit a revised version that addresses these concerns directly. Although we expect that you will address these comments in your response letter, we also need to see the corresponding revision in the text of the manuscript. Some of the reviewers' comments may seem to be simple queries or challenges that do not prompt revisions to the text. Please keep in mind, however, that readers may have the same perspective as the reviewers. Therefore, it is essential that you attempt to amend or expand the text to clarify the narrative accordingly.

Given the present situation, we will give authors as much time as they need to submit revised manuscripts. We are also offering, if you choose, to post the manuscript to bioRxiv (if it is not already there) along with this decision letter and a formal designation that the manuscript is "in revision at *eLife*". Please let us know if you would like to pursue this option. (If your work is more suitable for medRxiv, you will need to post the preprint yourself, as the mechanisms for us to do so are still in development.)

Summary:

Sense of coherence (SoC) refers to a person's psychological ability to properly adapt to stress and adversity during the life course. The scale was developed to measure how people view life and to identify how a person may use her/ his intrinsic resources to overcome difficulties while maintaining their health. It seeks to understand why some people remain healthy under stress while others become sick. The SoC concept has been widely used and there is evidence suggesting a link between SoC and several health outcomes including cardiovascular disease and diabetes. However, the association between SoC and cancer is less studied. In the current study, the authors examine the association between SoC and risk of breast cancer using data from a cohort study of 46,436 women recruited in the Karolinska Mammography Project for Risk of Breast Cancer (Karma). This well-conducted study did not show an association between SoC and breast cancer. This study represents a potentially valuable contribution to the literature on SoC and breast cancer. However, there are a few substantive concerns.

Essential revisions:

While the authors provide a brief explanation of the SoC, the theoretical framework is still underdeveloped, and the explanations and definitions are vague and unclear. The text would benefit from a more elaborated explanation of theory of salutogenesis and how it relates to breast cancer. In addition, the authors need to provide a more comprehensive definition of SoC and its three aspects of comprehensibility, manageability and meaningfulness, notably explaining more about the nature of the stimuli as mentioned in the paper. A clearer definition of Generalized Resistance Resources (GRRs) is also required.

In summary, the manuscript fails to highlight the theoretical importance of examining the link between SoC and breast cancer, except that previous studies' results have been mixed or that no clear relationship has been established for this association. It is unclear to the reviewer which aspect of this current manuscript can resolve the issues of the existing mixed results on this topic.

The analytical approach of the paper needs to be improved; we recommend that the authors run a sensitivity analysis without adjustment for life stressors, hours of sleep, and quality of life. This is for the following reasons:

• "Sense of coherence (SOC) reflects a coping capacity of people to deal with everyday life stressors". Therefore, life stressors should not be adjusted for; the effects of SOC are likely washed away when stressful life events are included as confounders (over adjustment).

• Having a high SOC means that one is able to cope better with stressors in the surrounding environment; therefore, how stressed one feels is an inherent part of SOC and should not be adjusted for.

• Quality of life (QOL) is also typically assessed as an outcome of SOC, rather than as a confounder as this paper has done. The stronger the SOC, the better the quality of life (https://www.ncbi.nlm.nih.gov/pmc/articles/PMC2465600/). Please omit QOL from the confounder list in the sensitivity analysis.

• The study by Poppius et al. cited in the Discussion section did find an association with cancer in the shorter follow-up period of 8 years. In contrast to this study on SOC and breast cancer, Poppius et al. only adjusted for age, occupation, smoking, and alcohol consumption. In line with the above comments, the authors of the current study are over adjusting. Also, the Poppius et al. study combined all cancers and measured their association with SOC, so the endpoint is different. Hence, the differences in results between studies.

According to the SOC literature: "Generalized Resistance Resources (GRR) comprise the characteristics of a person, a group, or a community that facilitate the individuals' abilities to cope effectively with stressors and contribute to the development of the individual's level of (SOC)"

Table 1 – GRRs are part of the SOC concept, they shouldn't be separated from it (the authors should not assess the relationship between GRRs and SOC, because the former is part of the latter). Please change the title of Table 1 to "participant characteristics associated with SOC" or something along those lines. Better to show that you are assessing the relationship of SOC with sociodemographics, lifestyle factors, etc. (potential confounders), and dispense with GRR terminology.

---

## [Author Response]

Essential revisions:While the authors provide a brief explanation of the SoC, the theoretical framework is still underdeveloped, and the explanations and definitions are vague and unclear. The text would benefit from a more elaborated explanation of theory of salutogenesis and how it relates to breast cancer. In addition, the authors need to provide a more comprehensive definition of SoC and its three aspects of comprehensibility, manageability and meaningfulness, notably explaining more about the nature of the stimuli as mentioned in the paper. A clearer definition of Generalized Resistance Resources (GRRs) is also required.

Thank you for the comments. We have now (1) elaborated the explanation of the theory of salutogenesis and how it might relate to breast cancer, (2) provided a more comprehensive definition of SoC and its three components as well as explained more about the nature of the stimuli mentioned in the paper, and (3) provided a clearer definition of Generalized Resistance Resources.

1)

Explanation of the theory of salutogenesis and how it might relate to breast cancer.

Modified text (Introduction, paragraph one):

“The theory of salutogenesis was developed by Antonovsky (Antonovsky, 1979) to study factors that support human health and well-being, in contrast to “pathogenesis” which studies factors that cause diseases, with a specific focus on understanding the relationship between health, stress, and coping. According to this theory, health is a lifelong and dynamic process of actively coping with a variety of stress factors one might encounter in life (i.e., stimuli), and, although stress is ubiquitous, people experience different health outcomes in response to stress.”

Modified text (Introduction, paragraph three):

“The theory of salutogenesis has been applied in different fields of health and medicine. […] No study has addressed this research question among women and for female cancer.”

2)

Definition of SoC and its three components as well as explanation about the nature of the stimuli mentioned in the paper.

Modified text (Introduction, paragraphs one and two):

“According to this theory, health is a lifelong and dynamic process of actively coping with a variety of stress factors one might encounter in life (i.e., stimuli), and, although stress is ubiquitous, people experience different health outcomes in response to stress.

[…] Accordingly, SoC has three components, namely comprehensibility, manageability and meaningfulness, and salutogenesis is greatly dependent on a strong SoC."

3)

Definition of Generalized Resistance Resources.

Modified text (Introduction, paragraph two):

“Generalized Resistance Resources are one of the key determinants for SoC, consisting all resources that help a person cope and are effective in avoiding or combating a range of stress factors, including socioeconomic standing and social support (Antonovsky, 1972; Eriksson and Al-Yagon, 2016).”

In summary, the manuscript fails to highlight the theoretical importance of examining the link between SoC and breast cancer, except that previous studies' results have been mixed or that no clear relationship has been established for this association. It is unclear to the reviewer which aspect of this current manuscript can resolve the issues of the existing mixed results on this topic.

We have now highlighted the theoretical importance of examining the link between SoC and breast cancer, as well as the added value of our study to the existing results on this topic.

Modified text (Introduction, paragraph three):

“The theory of salutogenesis has been applied in different fields of health and medicine. […] To this end, we took advantage of a mammography screening cohort in Sweden and assessed the association of SoC with the subsequent risk of breast cancer.”

Modified text (Discussion, paragraph three):

“Poppius et al. (Poppius et al., 2006) reported a 52% higher risk of any cancer during an 8-year follow-up of middle-aged men in Finland, when comparing men with a weak SoC to men with a strong SoC. […] The role of SoC in other cancer types among women, in populations outside the Nordic countries, or among other age groups needs to be further investigated.”

The analytical approach of the paper needs to be improved; we recommend that the authors run a sensitivity analysis without adjustment for life stressors, hours of sleep, and quality of life. This is for the following reasons:• "Sense of coherence (SOC) reflects a coping capacity of people to deal with everyday life stressors". Therefore, life stressors should not be adjusted for; the effects of SOC are likely washed away when stressful life events are included as confounders (over adjustment).• Having a high SOC means that one is able to cope better with stressors in the surrounding environment; therefore, how stressed one feels is an inherent part of SOC and should not be adjusted for.• Quality of life (QOL) is also typically assessed as an outcome of SOC, rather than as a confounder as this paper has done. The stronger the SOC, the better the quality of life (https://www.ncbi.nlm.nih.gov/pmc/articles/PMC2465600/). Please omit QOL from the confounder list in the sensitivity analysis.• The study by Poppius et al. cited in the Discussion section did find an association with cancer in the shorter follow-up period of 8 years. In contrast to this study on SOC and breast cancer, Poppius et al. only adjusted for age, occupation, smoking, and alcohol consumption. In line with the above comments, the authors of the current study are over adjusting. Also, the Poppius et al. study combined all cancers and measured their association with SOC, so the endpoint is different. Hence, the differences in results between studies.

We agree with the reviewer that our original model 3 with the adjustment of life stressors, hours of sleep, stressed feeling and quality of life is likely overly conservative. We have now deleted model 3 from Table 2 and used model 2 as the full model in the results presented in Figure 2 and Table 3. We have also modified the text of the manuscript to reflect such changes.

Modified text (Materials and methods, paragraph three):

“In this study, we used information from the baseline questionnaire, including sociodemographic factors (age, ancestry, education), lifestyle factors (body mass index [BMI], smoking, drinking, physical activity), and other known risk factors for breast cancer including reproductive and hormonal factors, family history of breast cancer, history of benign breast disorders or other malignancies. We also used breast density percent measured at baseline by previously reported STRATUS software (Lindblad et al., 2018).”

Modified text (Materials and methods, paragraph four):

“We first fitted a model not adjusted for any other covariables (Model 1). In a full model (Model 2), we adjusted the analyses for sociodemographic factors (age, ancestry, and education), lifestyle factors (body mass index, smoking, alcohol consumption, and physical activity), and known risk factors for breast cancer including reproductive and hormonal factors (age at menarche, age at first birth, No. of pregnancies, No. of births, use of contraceptives, use of hormone replacement therapy, menopausal status), as well as family history of breast cancer, benign breast disorders, other malignancies, and breast density.”

*“*We further performed a sub-analysis to assess whether the studied association would differ between subtypes of breast cancer using the full model.”

Modified text (Results, paragraph one):

“More than half of the women were post-menopausal, and had college education or above (Table 1). Women with strong SoC were slightly older, more educated, less likely to smoke, and more physically active.”

Finally, we have also modified the text that are relevant to the Poppius et al. study in the Introduction and Discussion sections to illustrate the potential explanations for the different findings observed between the Poppius et al. study and the present study.

Modified text (Introduction, paragraph three):

“In contrary, there is so far one study, to the best of our knowledge, that assessed the role of SoC on cancer development, and found an association between weak SoC and an increased risk of cancer during an 8-year follow-up, although not 12-year follow-up, of a population of middle-aged Finnish men (Poppius et al., 2006). No study has addressed this research question among women and for female cancer.”

Modified text (Discussion, paragraph three):

“To our knowledge, few studies have examined the link between SoC and risk of cancer. Poppius et al. (Poppius et al., 2006) reported a 52% higher risk of any cancer during an 8-year follow-up of middle-aged men in Finland, when comparing men with a weak SoC to men with a strong SoC. […] The role of SoC in other cancer types among women, in populations outside the Nordic countries, or among other age groups needs to be further investigated.”

According to the SOC literature: "Generalized Resistance Resources (GRR) comprise the characteristics of a person, a group, or a community that facilitate the individuals' abilities to cope effectively with stressors and contribute to the development of the individual's level of (SOC)"Table 1 – GRRs are part of the SOC concept, they shouldn't be separated from it (the authors should not assess the relationship between GRRs and SOC, because the former is part of the latter). Please change the title of Table 1 to "participant characteristics associated with SOC" or something along those lines. Better to show that you are assessing the relationship of SOC with sociodemographics, lifestyle factors, etc. (potential confounders), and dispense with GRR terminology.

We agree with the reviewer(s) and have removed results about life stressors, stressed feeling, hours of sleep, and quality of life from Figure 1 and changed the title of Figure 1 as “Linear associations between sociodemographic and lifestyle factors and sense of coherence”. We have also removed these items from Table 1 and changed the title of Table 1 as “Characteristics of study participants by sense of coherence at baseline”. We have also modified the text of the manuscript to reflect such changes.

Modified text (Materials and methods, paragraph three):

“In this study, we used information from the baseline questionnaire, including sociodemographic factors (age, ancestry, education), lifestyle factors (body mass index [BMI], smoking, drinking, physical activity), and other known risk factors for breast cancer including reproductive and hormonal factors, family history of breast cancer, history of benign breast disorders or other malignancies. We also used breast density percent measured at baseline by previously reported STRATUS software (Lindblad et al., 2018).”

Modified text (Results, paragraph two):

“In the linear regression, we found SoC to be positively associated with age (coefficient=0.13, 95% CI: 0.12 to 0.14), number of births (coefficient=0.49, 95% CI: 0.39 to 0.59), and a higher physical activity (coefficient=1.24, 95% CI: 0.98 to 1.5 for high vs. low physical activities) (Figure 1). Higher BMI (coefficient=-0.17, 95% CI: -0.19 to -0.14), non-European ancestry (coefficient=-5.15, 95% CI: -5.8 to -4.5), less education (coefficient=-2.13, 95% CI: -2.46 to -1.8 for ≤ 10 years vs. >12 years of education), previous or current smoking (coefficient=-0.99, 95% CI: -1.22 to -0.77 for previous smoker vs. non-smoker; coefficient=-2.45, 95% CI: -2.78 to -2.12 for current smoker vs. non-smoker), and never or ≥ 10g/d drinking (coefficient=-2.23, 95% CI: -2.51 to -1.95 for non-drinker vs. <10 g/d of alcohol consumption; coefficient=-0.63, 95% CI: -0.9 to -0.36 for ≥ 10 g/d vs. <10 g/d of alcohol consumption) were all associated with a lower SoC score.”